# OpenReview forum: "AlphaSteer: Learning Refusal Steering with Principled Null-Space Constraint"
_ICLR.cc/2026/Conference — ICLR 2026 Poster_

### Official Review · Reviewer_4M51 · 2025-10-28

**Soundness:** 3
**Presentation:** 4
**Contribution:** 3
**Rating:** 8
**Confidence:** 4

**Summary:**

This paper proposes a simple, effective, and principled method called AlphaSteer, which steers the activations of LLMs to refuse malicious prompts while retaining maximum utility for benign ones. Specifically, AlphaSteer defines an explicit objective for this goal and derives an efficient approach to achieve it without exhaustively retraining model parameters for safety alignment. The experimental results demonstrate the effectiveness of the proposed method.

**Strengths:**

- The presentation is clear and easy to follow.

- The idea is simple yet principled: the goal of this paper is rigorously defined, and the proposed approach to achieve it is both efficient and well-justified. In particular, introducing the concept of having zero effect on benign prompts (rather than explicitly maximizing utility, such as the log-likelihood of outputs) is a reasonable formulation.

- The experimental results are strong, at least within the scope of the setups presented in this paper.

**Weaknesses:**

I think this paper is already strong, but the following points could further improve it:

- The proposed method appears lightweight (mainly involving SVD computation and matrix multiplication in a full-batch manner). However, in my view, it is still data-driven. It would therefore be helpful to compare this approach with a fully data-driven baseline — for example, a simple supervised fine-tuning model trained to generate refusals for malicious prompts in the same dataset $\mathcal{D}_m$. Although such a baseline might overfit $\mathcal{D}_m$, it would still highlight the advantages of the proposed method. Even if the baseline performs better, AlphaSteer would remain preferable due to its efficiency.

- AlphaSteer introduces some additional computational overhead (which appears marginal), but it would be useful to discuss this overhead in more detail — particularly in comparison to the baseline (i.e., only computing the refusal vector $r$).

- In certain cases (e.g., Llama-3.1-8B-Instruct on Math and GSM8K), AlphaSteer actually improves utility. This suggests that AlphaSteer might have a regularization effect (e.g., the input $h_b$ being influenced by $\tilde{\delta}$ when moving out of the null space). Providing intuition or analysis for this phenomenon could further support the claim that AlphaSteer enhances both safety and utility.

- The paper studies the effect of the steering strength $\lambda$ in Figure 11. Could an optimal $\lambda$ be derived using a similar objective formulation?

**Questions:**

See the weaknesses

---

> ### Author Response · Authors · 2025-11-23
> **Official Response - Part 1**
>
> We sincerely appreciate your positive feedback and valuable suggestions. We respond to your questions as follows:
>
> ---
>
> > **Comparison with data-driven baselines**
>
> Thanks for the suggestion. We compared the proposed AlphaSteer and data-driven baseline supervised fine-tuning in Appendix D.2, where we report the performance together with the efficiency comparison again as follows:
>
> **Table 1: Performance comparison on safety enhancement with SFT**
>
> |                       |   AIM   | AutoDAN | Cipher |  GCG   | Jailbroken |  PAIR  | ReNeLLM |  Avg DSR  |
> | :- | :-: | :-: | :-: | :-: | :-: | :-: | :-: | :-: |
> | Llama-3.1-8B-Instruct |   92    |   48    |   0    |   58   |     75     |   45   |   28    |   48.00   |
> | + SFT                 |   100   |   97    |   31   | **99** |     81     |   48   |   24    |   68.57   |
> | + AlphaSteer          | **100** | **99**  | **63** |   97   |   **92**   | **98** | **100** | **91.93** |
>
> **Table 2: Performance comparison on utility preservation with SFT**
>
> |                       |  XSTest  | AlpacaEval |   MATH   |  GSM8K   | Avg Score |
> | :- | :-: | :-: | :-: | :-: | :-: |
> | Llama-3.1-8B-Instruct |   92.4   |    50.0    |   45.0   |   81.0   |   67.1    |
> | + SFT                 |   90.0   |    31.4    |   27.0   |   79.0   |   56.9    |
> | + AlphaSteer          | **91.2** |  **48.1**  | **46.0** | **84.0** | **67.3**  |
>
> As shown above, our proposed method not only achieves better performance on both safety enhancement and utility preservation.
>
> Additionally, we further compare our method with one more data-driven steering baseline WAS [1], which trains a tiny controller on LLM activations from harmful vs. benign prompts so it learns to predict per-layer scaling factor along the refusal direction. We report the performance comparison in Table 3 and 4. As shown in these Tables, AlphaSteer consistently outperforms this data-driven steering baseline. We attribute the success to the principled design of AlphaSteer.
>
> **Table 3: Performance comparison on safety enhancement**
>
> |Model|AIM|AutoDAN|Cipher|GCG|Jailbroken|PAIR|ReNeLLM|Avg DSR|
> |-|-|-|-|-|-|-|-|-|
> |**Llama-3.1-8B-Instruct**|92|48|0|58|75|45|28|48.00|
> |**+ WAS**|92|46|17|99|76|47|24|57.29|
> |**+ AlphaSteer**|100|99|63|97|92|98|100|91.93|
>
> **Table 4: Performance comparison on utility preservation**
>
> ||XSTest|AlpacaEval|MATH|GSM8K|Avg Score|
> |-|-|-|-|-|-|
> |**Llama-3.1-8B-Instruct**|92.4|50.0|45.0|81.0|67.1|
> |**+ WAS**|92.0|49.7|44.0|80.0|66.4|
> |**+ AlphaSteer**|91.2|48.1|46.0|84.0|67.3|
>
> [1] Guiding Giants: Lightweight Controllers for Weighted Activation Steering in LLMs. Arxiv. 2025.5

---

> ### Author Response · Authors · 2025-11-23
> **Official Response - Part 2**
>
> > **Computational Cost**
>
> Thanks for your suggestions. Adding a discussion about the computational costs will further help us understand the practical usage of our method. We discuss the computational cost from the complexity and run-time cost perspectives.
>
> For computational complexity, we discuss it as follows:
>
> Let $L$ be the total number of model layers, $D$ the embedding dimension, $N_b$ the number of benign samples, $N_m$ the number of malicious samples, and $0 < p < 1$ the proportion of retained singular vectors. The per-layer computation is outlined in the table below, with the overall complexity being $O(L (N_b D^2 + N_m D^2 + D^3))$.
>
> | Step | Description | Complexity |
> |:-|:-:|-|
> | 1 | Compute $\mathbf{H}_b \mathbf{H}_b^\top$ | $O(N_b D^2)$ |
> | 2 | Perform SVD: $\mathbf{H}_b \mathbf{H}_b^\top = \mathbf{U} \mathbf{\Lambda} \mathbf{U}^\top$ | $O(D^3)$ |
> | 3 | Select $p \cdot D$ zero-eigenvalue vectors to form $\hat{\mathbf{U}}$, compute $\hat{\mathbf{P}} = \hat{\mathbf{U}} \hat{\mathbf{U}}^\top$ | $O(p D^3)$ |
> | 4 | Compute $\mathbf{X} = \mathbf{H}_m \hat{\mathbf{P}}$ | $O(N_m D^2)$ |
> | 5 | Compute $\mathbf{A} = \mathbf{X}^\top \mathbf{X} + \alpha (\hat{\mathbf{P}}^\top \hat{\mathbf{P}})$ | $O(N_m D^2 + D^3)$ |
> | 6 | Compute $\mathbf{b} = \mathbf{X}^\top (\mathbf{r} \cdot \text{repeat}(N_m))$ | $O(N_m D^2)$ |
> | 7 | Compute $\tilde{\mathbf{\Delta}} = \mathbf{A}^+ \mathbf{b}$ | $O(D^3)$ |
> | 8 | Compute $\mathbf{\Delta} = \tilde{\mathbf{\Delta}} \hat{\mathbf{P}}$ | $O(D^3)$ |
>
> For run-time cost, we compare the total cost of training with supervised fine-tuning under the same amount of training data in Table 4. As shown in this table, our method only introduces marginal training cost compared with SFT, since we do not need to train the whole LLM.
>
> **Table 5: Training time comparison**
>
> ||AlphaSteer|SFT|
> |-|-|-|
> |Time Cost|90s|20min|
>
> For inference costs, we compare AlphaSteer with baselines in Appendix C.2 due to the space limit.
>
> > **The effect of AlphaSteer on the math dataset**
>
> Thanks for your detailed observation. We would like to point out that, actually, only in very few cases that the performance on math improve, and in most cases, the performance remains almost the same. Therefore, we believe that it would be tricky for us to simply conclude that AlphaSteer can improve the mathematical performance of the model in a certain regularization way.
>
> > **The selection of $\lambda$**
>
> Thanks for your question. We believe this is one very interesting question for further exploration, but this is also hard and requires much more effort. We can have a rough sketch discussion about it here.
> To have such an estimation of the best $\lambda$, we need to first have a formal definition of it.
> For example, we can define it as a minimal DSR (e.g., 90%).
> Based on this definition, we can then calculate the mean disturbance on the activations of malicious data (e.g., the $\lambda$), together with the improvements on the DSR. Then we can establish an estimation function, where the independent variable and dependent variable are $\lambda$ and DSR, respectively. However, this can be hard to implement in practice, as we can observe in Figure 11 that the DSR does not always increase linearly as the $\lambda$ increases.
> We will try to study this question in our future work.

---

> > ### Comment · Reviewer_4M51 · 2025-11-27
> >
> > Thank you for your detailed response, which resolves many of my concerns.
> > I noticed that Appendix D.2 only summarizes jailbreak attacks, rather than comparing against the data-driven baseline supervised fine-tuning. Could you clarify this point?

---

> > > ### Author Response · Authors · 2025-11-28
> > > **Clarification**
> > >
> > > Thanks for the follow-up question. Actually, it is reported in Appendix E.2. Sorry for this typo.

---

### Official Review · Reviewer_kvZk · 2025-10-28

**Soundness:** 3
**Presentation:** 3
**Contribution:** 2
**Rating:** 4
**Confidence:** 4

**Summary:**

This paper proposes an activation steering method with a learnable refusal vector to defend against jailbreak attacks in LLMs. The learnable vector is optimized to balance the trade-offs between utility and safety. Experiments are carried out on three open-source LLMs with recent jailbreak attacks and utility benchmarks to show the effectiveness of the proposed defense.

**Strengths:**

- The proposed defense achieves  a better utility score (even slightly better than standard models on average).
- The paper shows theoretical grounding on its optimization of the learnable refusal vector.
- The proposed method achieves a better defense success rate on average against recent jailbreak attacks.
- The paper is well written and easy to read.

**Weaknesses:**

- The contribution may be limited as there are other existing learnable activation-steering methods considering before ICLR submission deadline.
The general learnable activation steering methods include:

[1] https://arxiv.org/abs/2505.20309v2 (version 1 released in May 2025)

[2] https://arxiv.org/abs/2506.03292 (hypernetwork-based steering)

[3] https://aclanthology.org/2024.findings-emnlp.479.pdf

The reviewer skips the paper after September 2025.

- The experiments are not rigorous. Better attacks, such as "do anything now" [a], AdvPrompter [b], are not used for evaluation.

[a]https://arxiv.org/abs/2308.03825

[b]https://arxiv.org/pdf/2404.16873

- Case study (RQ3) should be an in-depth analysis rather than showing an example of (ReNeLLM).
- The generalization ability of the learned refusal vector is not clearly explained, although there are experimental results on generalization without math data in the appendix (D.4).

Minor:
- The caption of Fig. 4 is missing.
- The small graphs in the supplementary materials are not readable.

**Questions:**

- Activation steering is known to introduce safety and alignment risks. How does the proposed method guarantee not to introduce other safety and alignment risks other than jailbreak attacks at hand?

- The steering vector may not generalize well beyond the defined settings or prompt types. What is the expected generalization?

- How does the proposed method guarantee the learned steering direction is reliable? (Fidelity)

- The design of the prompts may affect the steering direction. What is the variance?
How $D_b$ and $N_m$ are constructed?

- The limitations say the effectiveness is unknown for large reasoning models. What about small reasoning models such as Phi-3?

---

> ### Author Response · Authors · 2025-11-23
> **Official Response - Part 1**
>
> We sincerely appreciate your suggestions, which can greatly help us to refine our paper. We respond to your questions as follows:
>
> ---
>
> > **W1: Discussion and comparison with more baselines**
>
> Thanks for bringing up the discussion on relevant activation steering methods. We would like to highlight that our method is principled and theoretically guided. Therefore, the learnable steering mechanism we proposed is inseparable and complementary to our principled design. Such a design distinguishes our work from the existing methods, since existing works still largely lack principled guidance. We also add discussion about such papers in the related work part in Appendix A.2. Since only the WAS [1] is designed for safety enhancement, we additionally add comparison experiments with it, and report the performance results in Tables 1 and 2. The results show that AlphaSteer also outperforms this baseline.
>
> **Table 1: Performance comparison on safety enhancement. **
>
> |Model|AIM|AutoDAN|Cipher|GCG|Jailbroken|PAIR|ReNeLLM|Avg DSR|
> |-|-|-|-|-|-|-|-|-|
> |**Llama-3.1-8B-Instruct**|92|48|0|58|75|45|28|48.00|
> |**+ WAS**|92|46|17|99|76|47|24|57.29|
> |**+ AlphaSteer**|100|99|63|97|92|98|100|91.93|
>
> **Table 2: Performance comparison on utility preservation. **
>
> ||XSTest|AlpacaEval|MATH|GSM8K|Avg Score|
> |-|-|-|-|-|-|
> |**Llama-3.1-8B-Instruct**|92.4|50.0|45.0|81.0|67.1|
> |**+ WAS**|92.0|49.7|44.0|80.0|66.4|
> |**+ AlphaSteer**|91.2|48.1|46.0|84.0|67.3|
>
> [1] Guiding Giants: Lightweight Controllers for Weighted Activation Steering in LLMs. Arxiv. 2025.5
>
> > **W2: Consideration of more jailbreak attacks**
>
> Thanks for the suggestion. For the "do anything now" (DAN) attack, we have compared a more advanced version of it, AutoDan, which constructs a better attack with dynamic adjustment. To address this concern, we add more experiments of our method on AdvPrompt [2], FlipAttack [3], and WildGuard Test [4]. We do not train our method again and only conduct inference on these attacks. We report the performance of our method in Table 3.
> After applying AlphaSteer with the default strength 0.5, the model achieves near-perfect robustness across all benchmarks: 100% on AdvPrompter, 84–100% on most FlipAttack modes, and 100% on WildGuard Test.
> These results show the strong generalization capabilities of AlphaSteer on diverse malicious scenarios.
> More details can be found in Appendix E.5.
>
> **Table 3: Llama-3.1-8B-Instruct with AlphaSteer on more safety benchmarks**
>
> | Strength       | 0 (original) | 0.1 | 0.2 | 0.3 | 0.4 | 0.45 | 0.5 |
> |----------------|--------------|-----|-----|-----|-----|------|-----|
> | AdvPrompter    | 61| 99  | 100 | 100 | 100 | 100  | 100 |
> | FlipAttack-FCS | 0| 0   | 0   | 0   | 2   | 49   | 99  |
> | FlipAttack-FCW | 0| 0   | 0   | 0   | 0   | 18   | 84  |
> | FlipAttack-FMM | 0| 0   | 0   | 0   | 15  | 88   | 100 |
> | FlipAttack-FWO | 2| 2   | 17  | 65  | 100 | 100  | 100 |
> | WildGuard Test | 70.5| 82.6| 94.2| 98.1| 99  | 100  | 100 |
>
> [2] AdvPrompter: Fast Adaptive Adversarial Prompting for LLMs.
>
> [3] FlipAttack: Jailbreak LLMs via Flipping.
>
> [4] WildGuard: Open One-Stop Moderation Tools for Safety Risks, Jailbreaks, and Refusals of LLMs.
>
> > **W3: In-depth analysis of the case study**
>
> We conduct an in-depth analysis of the case study from the perspective of activations to illustrate how AlphaSteer succeeds while the vanilla model fails. Specifically, for the activation of the malicious prompt, it is steered towards refusal. For the activation of the benign prompt, it remains largely unchanged. More details and visualization can be found in Appendix F.2.
>
> > **W4: Explanation about the generalization capabilities of math data**
>
> We try to explain the generalization capabilities of the AlphaSteer from the perspective of space coverage. We introduce the concept of space coverage in Appendix F.7. It measures whether the activations in the test set lie in the space that the benign activations in the train set span. Specifically, we introduce the Projection‑Energy Coverage (PEC) [5] metric to evaluate the space coverage (more details can be found in Appendix F.7). The larger this value is, the better the utility preservation is. We calculate the average PEC score across all the math problems when keeping and removing relevant training data. The average PEC scores are 0.97 and 0.92, respectively, when math data is included and when all math data is removed.
> The relatively high PEC score of 0.92 after removing the math-related data indicates a still high space coverage. Therefore, the performance on math data is largely unaffected. More details can be found in Appendix E.4.
>
> [5] Sparse subspace clustering: Algorithm, theory, and applications. TPAMI 2013.
>
>
> > **W5: Minor**
>
> Thanks for the detailed evaluation. We have fixed Figure 4. For small graphs, it is hard to increase the font size since we have a lot of sub-figures within one main figure. It would be readable when zooming up.

---

> ### Author Response · Authors · 2025-11-23
> **Official Response - Part 2**
>
> > **Q1: Other possible safety risks**
>
> Thanks for the question. I may not have fully understood this question, and I'm willing to discuss it further if my understanding is incorrect. It is true that activation steering may introduce safety and alignment risks if we do not properly add constraints on it, and this is also why these baseline methods, such as vector calibration and conditional steering, tend to fail in several cases. In contrast, in our method, we apply principled null space constraints on benign prompts to minimize the risks of these data. This principled design minimizes the possibility of introducing additional safety risks.
>
> > **Q2: The expected generalization**
>
> As we have reported in our response to W2, our method generalizes well to more diverse scenarios. This is because the refusal vector used in our paper actually reflects the refusal behavior and can consistently yield refusal behaviors, and our principled design ensures that the benign activations are largely unaffected.
>
> > **Q3 & Q4: The reliability of the steering direction & the Design of prompts for constructing the steering direction**
>
> Thanks for the question. The main contribution of this paper does not lie in finding a better steering direction vector. Instead, we just follow the well-defined framework of constructing the refusal vector in previous work [6], which is proven to consistently yield refusal behaviors.
> For the discussion of the variance, we think the research line of vector calibration can be taken into consideration.
> In these works, they try to reduce the effect on benign data through editing the steering vector and finally deliver sub-optimal performance. Such special calibration on vectors leads to the variance in the quality of these vectors for inducing refusal.
> Instead, the refusal vector in our paper is extracted by conducting mean-in-difference on malicious prompts that induce refusal and compliance behaviors, respectively, which provide strong refusal capabilities as shown in the ablation study of RV. This indicates the reliability of the steering direction used in our paper.
>
> [6] Refusal in Language Models Is Mediated by a Single Direction. NeurIPS 2024.
>
> > **Q5: Effectiveness on small models**
>
> Thanks. We have reported the performance on two small-sized models (i.e., Llama-3.2-1B-Instruct and Llama-3.2-3B-Instruct). Our proposed method also works fine on these small-sized models. We report the performance as follows again in Tables 4 and 5. It is shown that AlphaSteer also performs well on these models.
>
> **Table 4: Performance comparison on safety enhancement**
>
> |                           | AutoDan | Cipher | Jailbroken | PAIR | ReNeLLM | WildGuardTest | Avg DSR |
> | :------------------------ | :--: | :-----: | :--: | :--------: | :--: | :-----: | :-----: |
> | **Llama-3.2-1B-Instruct** |  29  |   34   |  82  |     87     |  27  |   90.8   |  58.30  |
> | + AlphaSteer              | 94 |   97    |  99  |     100     |  97  |   98.8   |  97.63  |
> | **Llama-3.2-3B-Instruct** |   53    |   47   |     86     |  77  |   46    |     68.4      |  62.90  |
> | + AlphaSteer              |   99    |   76   |     98     |  99  |   97    |     95.7      |  94.11  |
>
> **Table 5: Performance comparison on utility preservation**
>
> |                           | XSTest | AlpacaEval | MATH | GSM8K | Avg Score |
> | :------------------------ | :----: | :--------: | :--: | :---: | :-------: |
> | **Llama-3.1-1B-Instruct** |  84.4  |    50.0    | 22.0 | 26.0  |   45.60   |
> | + AlphaSteer              |  81.6  |    49.8    | 20.0 | 24.0  |   43.85   |
> | **Llama-3.2-3B-Instruct** |  96.8  |    50.0    | 35.0 | 73.0  |   63.70   |
> | + AlphaSteer              |  94.4  |    50.1    | 37.0 | 71.0  |   63.13   |

---

> > ### Comment · Reviewer_kvZk · 2025-11-24
> >
> > Thank you for addressing all my concerns.
> > Regarding **Q1: Other possible safety risks**,
> >
> > The previous work showed that steering for some concept affects the safety [1].
> > It seems alphasteer takes care of the side effects with the specific constraints.
> > Thank you for the explanation. I have increased the score.
> >
> > [1] https://arxiv.org/pdf/2509.22067v1

---

> > > ### Author Response · Authors · 2025-11-24
> > > **Thanks!**
> > >
> > > Thanks so much for your timely reply and positive feedback! We are glad that our response can address your concerns.

---

### Official Review · Reviewer_V3rz · 2025-10-28

**Soundness:** 3
**Presentation:** 4
**Contribution:** 4
**Rating:** 8
**Confidence:** 3

**Summary:**

AlphaSteer introduces a learnable activation-steering method that keeps a model’s normal behavior intact while strengthening its tendency to refuse harmful requests. It first carves out a space that represents benign behavior and minimizes any steering there, then learns an adaptive “refusal direction” from activation data so the model gently shifts toward safe responses only when prompts are malicious. Across multiple open instruction models and a range of common jailbreak attacks, it raises defense success while largely preserving compliance and standard task performance, outperforming prior refusal-vector baselines.

**Strengths:**

1. The method grounds activation steering in a clear linear-algebraic framework: (1) preserve utility by projecting benign activations into a learned (near) null-space, and (2) enhance safety via an adaptive, data-driven refusal vector estimated in closed form.
2. Across diverse jailbreak families, the approach delivers state-of-the-art (SOTA) defense success on malicious prompts while maintaining (or minimally impacting) compliance and standard-task performance on benign prompts—consistently outperforming refusal-vector baselines and contemporary steering methods under comparable settings.
3. Clear geometry-focused visualizations (activation trajectories, norm-separation) and ablations (layer choice, steering strength, linear vs. MLP) justify each design choice and make the mechanism easy to audit and reproduce, strengthening both clarity and credibility.

**Weaknesses:**

1. The proposed method includes introduction of the computation of null-space projection matrix, but does not show whether the new computation is costly. For showing effective practical usage, it would help to compare computation with existing baselines. For example, Surgical [1] offers Inference time and Memory comparison.
2. The evaluation solely depends on GPT-4o model as LLM-for-judge for DSR (Defense Success Rate) and CR (Compliance Rate), while having no justification for the model selection. Although it is based on GPT-4 not GPT-4o, WIldGuard[2] shows that guard-specific models can serve as better judges. You might want to include other guard-specific models as independent judges, and report how the results change for further validation.


[1] Wang, X., Hu, C., Röttger, P., & Plank, B. (2024). Surgical, cheap, and flexible: Mitigating false refusal in language models via single vector ablation. arXiv preprint arXiv:2410.03415.

[2] Han, S., Rao, K., Ettinger, A., Jiang, L., Lin, B. Y., Lambert, N., ... & Dziri, N. (2024). Wildguard: Open one-stop moderation tools for safety risks, jailbreaks, and refusals of llms. Advances in Neural Information Processing Systems, 37, 8093-8131.

**Questions:**

1. Please state more details about the content and intent deduplication method in C.1.

---

> ### Author Response · Authors · 2025-11-23
> **Official Response**
>
> We highly appreciate your positive feedback and valuable suggestions about our paper. We respond to your questions as follows:
>
> ---
>
> > **The computational cost comparison.**
>
> Thanks for your suggestions about discussing the computational cost. We discuss this from three aspects: training computational complexity, training time in practice, and inference costs.
>
> For computational complexity, we discuss it as follows:
>
> Let $L$ be the total number of model layers, $D$ the embedding dimension, $N_b$ the number of benign samples, $N_m$ the number of malicious samples, and $0 < p < 1$ the proportion of retained singular vectors. The per-layer computation is outlined in the table below, with the overall complexity being $O(L (N_b D^2 + N_m D^2 + D^3))$.
>
> **Table 1: Training time comparison**
>
> | Step | Description | Complexity |
> |:-|:-:|-|
> | 1 | Compute $\mathbf{H}_b \mathbf{H}_b^\top$ | $O(N_b D^2)$ |
> | 2 | Perform SVD: $\mathbf{H}_b \mathbf{H}_b^\top = \mathbf{U} \mathbf{\Lambda} \mathbf{U}^\top$ | $O(D^3)$ |
> | 3 | Select $p \cdot D$ zero-eigenvalue vectors to form $\hat{\mathbf{U}}$, compute $\hat{\mathbf{P}} = \hat{\mathbf{U}} \hat{\mathbf{U}}^\top$ | $O(p D^3)$ |
> | 4 | Compute $\mathbf{X} = \mathbf{H}_m \hat{\mathbf{P}}$ | $O(N_m D^2)$ |
> | 5 | Compute $\mathbf{A} = \mathbf{X}^\top \mathbf{X} + \alpha (\hat{\mathbf{P}}^\top \hat{\mathbf{P}})$ | $O(N_m D^2 + D^3)$ |
> | 6 | Compute $\mathbf{b} = \mathbf{X}^\top (\mathbf{r} \cdot \text{repeat}(N_m))$ | $O(N_m D^2)$ |
> | 7 | Compute $\tilde{\mathbf{\Delta}} = \mathbf{A}^+ \mathbf{b}$ | $O(D^3)$ |
> | 8 | Compute $\mathbf{\Delta} = \tilde{\mathbf{\Delta}} \hat{\mathbf{P}}$ | $O(D^3)$ |
>
> For the run-time training cost, we compare the total cost of training with supervised fine-tuning under the same amount of training data in Table 3. As shown in this table, our method only introduces marginal training cost compared with SFT, since we do not need to train the whole LLM.
>
> **Table 2: Training time comparison**
>
> ||AlphaSteer|SFT|
> |-|-|-|
> |Time Cost|90s|20min|
>
> We have added the details of the above analyses in Appendix C.2 and E.2, respectively.
>
> For inference costs, we compare AlphaSteer with baselines in Appendix C.2 due to the space limit.
>
>
>
>
> > **Evaluation with guard models**
>
> Thanks for your suggestions on evaluating with guard models. We add one more experiment of evaluating with Llama-Guard-3-8B. And we report the performance under this guard model in Table 3. Our proposed method, AlphaSteer, also performs well under the evaluation of Llama-Guard-3-8B, outperforming existing baselines.
> It is worth noting that Llama-Guard-3-8B tends to consistently yield higher DSR than GPT-4o, with a significant increase on Cipher. We analyze the difference between GPT-4o and Llama-Guard-3-8B and find out that Llama-Guard-3-8B tends to missclassify compliance responses as safe. More details can be found in Appendix G and Figure 22.
>
> **Table 3: Evaluation with different models**
>
> |Model|Evaluation Model|AIM|AutoDAN|Cipher|GCG|Jailbroken |PAIR|ReNeLLM|Avg DSR|
> |-|-|-|--|-|-|-|-|-|-|
> |**Llama3.1-8B-Instruct**|GPT-4o|92|48|0|58|75|45|28|48.00|
> ||Llama-Guard-3-8B|92|49|100|63|92|62|39|71.00|
> |**+ Jailbreak Antidote**  | GPT-4o | 100 | 97      | 0      | 100 | 86         | 93   | 63      | 76.94   |
> || Llama-Guard-3-8B|100|98| 100    | 96  | 97         | 98   | 71      |94.29|
> |**+ Surgical**|GPT-4o|100|76| 61     | 98  | 88         | 90   |67|82.83|
> || Llama-Guard-3-8B|100|78|100|96|98|94|78|92.00|
> | **+ CAST**|GPT-4o|92|51|67|99|81|96|96|80.57|
> |   | Llama-Guard-3-8B   | 92  | 52      | 100    | 98  | 95         | 97   | 97      | 90.14   |
> | **+ CircuitBreaker**      | GPT-4o | 100 | 100     | 34     | 100 | 80         | 96   | 81      | 84.42   |
> |   | Llama-Guard-3-8B   | 60  | 15      | 96     | 94  | 83         | 100  | 71      | 74.14   |
> | **+ Refusal SFT**         | GPT-4o | 100 | 97      | 31     | 99  | 81         | 48   | 24      | 68.57   |
> |   | Llama-Guard-3-8B   | 100 | 98      | 100    | 94  | 96         | 66   | 37      | 84.43   |
> | **+ RV**      | GPT-4o | 100 | 100     | 100    | 100 | 100        | 100  | 100     | 100.00  |
> |   | Llama-Guard-3-8B   | 100 | 100     | 100    | 100 | 100        | 100  | 100     | 100.00  |
> | **+ AlphaSteer**          | GPT-4o | 100 | 99      | 63     | 97  | 92         | 98   | 100     | 91.93   |
> |   | Llama-Guard-3-8B   | 100 | 99      | 100    | 98  | 98         | 100  | 100     | 99.29   |
>
>
> > **Details about the content and intent deduplication method in C.1**
>
> Thanks for your suggestions. We have added more details about the deduplication process in C.1.
> Specifically, we utilize the BLEU score to filter out all the evaluation prompts that have high scores between the malicious training prompts, following previous work Circuit Breaker [1], and more details can be found in our newly added sections in the paper.
>
> [1] Improving alignment and robustness with circuit breaker.

---

### Official Review · Reviewer_aqgX · 2025-11-01

**Soundness:** 4
**Presentation:** 3
**Contribution:** 4
**Rating:** 8
**Confidence:** 4

**Summary:**

This paper proposes AlphaSteer, a theoretically grounded activation steering method that addresses the safety-utility trade-off in LLM defenses against jailbreak attacks. Unlike conventional activation steering that indiscriminately applies refusal direction vectors to all prompts, AlphaSteer learns a transform matrix which produces steering vectors which are nearly zero for benign prompts (via null-space constraints for utility preservation) while maintaining refusal vectors for malicious prompts (via linear regression for safety enhancement). The method requires no additional post-training and demonstrates significant improvements in safety across multiple jailbreak attacks while maintaining general model capabilities.

**Strengths:**

- Strong theoretical foundation with principled learning objectives based on null-space constraints and linear regression, providing clear mathematical grounding for the approach.
- Addresses a critical limitation of existing activation steering methods with an elegant solution that treats benign and malicious prompts differently.
- Comprehensive experimental evaluation across multiple jailbreak attacks (GCG, AutoDAN, PAIR, etc.) and utility benchmarks demonstrating consistent improvements.
- Well-written with clear motivation and flow
- Strong results vs existing baselines

**Weaknesses:**

- The paper would benefit from more theoretical analysis of when and why the null-space constraint successfully preserves utility, and under what conditions it might fail.
- I think the paper would benefit from more details on how AlphaSteer is learned for the experiments to give a better sense of cost/scalability

**Questions:**

Does a transform matrix always have enough capacity to adequately learn when the difference between malicious and benign? Is AlphaSteer easy to trick if the attacker is aware ahead of time?

How does AlphaSteer perform against adaptive attacks where an adversary has knowledge of the learned steering vectors? Can the null-space constraints be circumvented by adversaries?

What is the computational overhead of learning AlphaSteer vs existing methods?

Under what conditions does the null-space constraint fail to preserve utility? Are there specific types of benign prompts that the authors observe still lose utility after AlphaSteer? How much is this affected by things like training set size.

[Figure 1] How is this plot created? By my understanding at this point in the paper, should the vanilla benign/malicious distributions be the same between Surgical and AlphaSteer? To me it looks like the benign vanilla distributions are different for surgical and alphasteer, why is that?

[98] Not a big deal, but it says recent studies and the first citation is from 1969.

[101] Extra space?

[366] This claim is too strong as Table 1 contricts the fact that 'AlphaSteer yields superior defense success rates across all the jailbreak attacks'

[Table 1 and 2] Can you discuss why you believe AlphaSteer underperforms on certain benchmarks/models compared to the baselines?

[411] The CAST papers claims that there is only a small increase in refusal rate for harmless prompts, can you explain why it is misclassifing math problems as malicious prompts, this seems surprising to me.

---

> ### Author Response · Authors · 2025-11-23
> **Official Response - Part 1**
>
> We sincerely appreciate your recognition of our work and the valuable suggestions! We respond to your comments as follows.
> Due to the page limit, most of our results are added in our latest paper.
>
> ---
>
> > **W1&Q4: More theoretical analysis, and under what conditions it might fail.**
>
> Following your suggestions, we further extend the analysis of the condition that AlphaSteer may fail.
> We point out that AlphaSteer may fail under two inappropriate training settings: insufficient benign training data, and inappropriate selection of the null-space ratio p%. More results are reported in our updated paper (Appendix E.7 and F.8.2).
>
> 1. When the benign training data is insufficient, the null space method can not faithfully cover the activation of test benign prompts, thereby inducing over-refusal.
>
> In Appendix E.7, we introduce a mathematical metric, Projection‑Energy Coverage (PEC) [1], for evaluating the coverage between the training and testing activation spaces, which is defined as $\mathrm{cov}_k(b) = \frac{|P_k b|_2^2}{|b|_2^2} \in [0,1]$. This metric examines whether the space that the training set's activation spans well covers the test set, where a larger PEC denotes better coverage and thereby reflects better utility preservation. Specifically, as shown in Table 1, when the space coverage is not enough for utility data (i.e., insufficient benign data is utilized for training), the proposed method tends to fail. We report more results together with the PEC analysis in Appendix E.7.1.
>
> **Table 1: Performance comparison on utility preservation**
>
> | Model                                      | XSTest | AlpacaEval | MATH | GSM8K | Avg Score |
> | ------------------------------------------ | ------ | ---------- | ---- | ----- | --------- |
> | Llama3.1-8B-Instruct                       | 92.4   | 50         | 45   | 81    | 67.10     |
> | with AlphaSteer (original 14k benign data) | 91.2   | 48.1       | 46   | 84    | 67.33     |
> | with AlphaSteer (3k benign data)           | 58     | 38.4       | 50   | 88    | 58.60     |
> | with AlphaSteer (1k benign data)           | 48     | 32.7       | 45   | 87    | 53.18     |
>
> 2. When the p% is too large, some of the column space is misclassified as the null space.
>
> Under this condition, the projection matrix can not always faithfully project the benign activations into the null space since some of the column space is misclassified as the null space. Therefore, AlphaSteer can not preserve the utility under this incorrect training setting. As shown in Table 2, when the p% is extremely large (i.e., p%=0.99), the utility drops dramatically.
> More details and results can be found in Appendix F.8.2.
>
> **Table 2: Performance comparison on utility preservation**
>
> |                           | XSTest | AlpacaEval | MATH | GSM8K | Avg Score |
> | :------------------------ | :----: | :--------: | :--: | :---: | :-------: |
> | **Llama-3.1-8B-Instruct** |  92.4  |    50.0    | 45.0 | 81.0  |   67.10   |
> | + AlphaSteer (p=0.1)      |  92.8  |    50.9    | 45.0 | 81.0  |   67.40   |
> | + AlphaSteer (p=0.3)      |  92.4  |    49.9    | 47.0 | 80.0  |   67.30   |
> | + AlphaSteer (p=0.6)      |  91.2  |    48.1    | 46.0 | 84.0  |   67.30   |
> | + AlphaSteer (p=0.9)      |  88.4  |    45.3    | 42.0 | 78.0  |   63.42   |
> | + AlphaSteer (p=0.99)     |  78.0  |    34.3    | 23.0 | 33.0  |   42.07   |
>
> [1] Sparse subspace clustering: Algorithm, theory, and applications. TPAMI 2013.
>
> > **W2: More details on how AlphaSteer is learned.**
>
> Thanks for your suggestions. We have added more training details in Appendix C.1. We also conduct one more computational complexity cost analysis in Appendix C.2 and the corresponding computational cost comparison with baselines.

---

> ### Author Response · Authors · 2025-11-23
> **Official Response - Part 2**
>
> > **Q1: Does a transform matrix always have enough capacity to adequately learn when the difference between malicious and benign?**
>
> As shown in Appendix C.1, we do not need a very large amount of malicious prompts to train the transform matrix, which only requires 2000 examples, to achieve a good performance. Therefore, under this condition, the transform matrix has enough capacity to learn from such a small amount of data and perform well on diverse scenarios.
>
> > **Q2: Performance against adaptive attacks.**
>
> Thanks for bringing up this important question. To examine whether our method can defend against adaptive attacks when the attacker knows our steering method in advance, we conduct the GCG attack against the LLM equipped with refusal vector and AlphaSteer, respectively. We report the performance of the GCG on the original model Llama-3.1-8B-Instruct and models equipped with refusal vector and AlphaSteer in Table 3. AlphaSteer maintains a high DSR against adaptive attacks. We attribute this to the principled design of AlphaSteer that makes such attacks hard to jailbreak. More details, including the training dynamics of GCG, can be found in the newly added Appendix E.6.
>
> **Table 3: Performance under adaptive attacks**
>
> ||DSR|Final Step Avg Loss|
> |-|-|-|
> |GCG on Llama-3.1-8B-Instruct|57.5|1.28|
> |GCG on Llama-3.1-8B-Instruct + RV Steer|79.5|1.55|
> |GCG on Llama-3.1-8B-Instruct + AlphaSteer|95.5|1.78|
>
> > **Q3: Computational overhead comparison.**
>
> Thanks for the suggestions. We have added the computational overhead comparison in the Appendix C.2.
>
> > **Q4: How is Figure 1 plotted.**
>
> We utilize the same vanilla data for drawing these figures. The distribution of the vanilla activations looks different because the post-steering activations are different, and the PCA visualization is based on all the activations (both before and after steering), which affects the visualization of vanilla activations.
>
> > **Q5,6,7: Format**
>
> Thanks for the detailed evaluation. For reference in line 98, the first reference is a mathematics book discussing the basic concepts of the null space. For line 101, we have removed the additional line. For line 366, we have revised it to avoid such strong claims.
>
> > **Q8: Performance on certain benchmarks.**
>
> We would like to highlight that AlphaSteer generally outperforms baselines on average, and only underperforms baselines on few benchmarks. The reason that these baselines yield slightly better performance on certain benchmarks can vary. For example, the training data used in these baseline methods are different from our method, which may introduce biases for specific benchmarks.
>
>
> > **Q9: Bad performance of CAST.**
>
> We add one detailed analysis about the bad performance of CAST in Appendix F.3.  CAST decides whether to apply steering by computing a scalar condition value, which is defined as the cosine similarity between the activation and a learned condition vector. It applies a refusal vector when this condition value falls below a threshold. However, when we apply CAST’s procedure to Llama-3.1-8B-Instruct, the condition values for MATH500/GSM8K prompts fall predominantly in the harmful-trigger region.
> In other words, CAST treats mathematical problems as “harmful-like” queries and therefore refuses to answer such questions.
> This phenomenon indicates the instability of such a heuristic design of the condition.

---

### Author Response · Authors · 2025-12-03
**Summary of Rebuttal**

Dear Reviewers and ACs,

We sincerely appreciate your time and efforts during the rebuttal phase. Your suggestions have greatly helped us refine our work.

In this work, we propose a theoretically grounded and empirically effective activation steering method called AlphaSteer, which introduces learnable activation steering mechanisms with null-space constraints. Experiments show the effectiveness of addressing the trade-off between safety and utility.

We are delighted that all reviewers highly recognize the proposed method, with scores of 8-8-8-8 as of Nov 24, after we effectively addressed all the concerns raised by reviewer kvZk. Specifically, all the reviewers (reviewers $\color{red}{\text{aqgX}}$, $\color{blue}{\text{V3rz}}$, $\color{green}{\text{kvZk}}$, $\color{orange}{\text{4M51}}$) appreciated the principled design and theoretical grounding, the well-organized writing, and the strong results. Additionally, some reviewers also highlighted the simplicity (reviewer $\color{orange}{\text{4M51}}$), the clear visualizations (reviewer $\color{blue}{\text{V3rz}}$), and the elegance of the method (reviewer $\color{red}{\text{aqgX}}$). The reviewers also raised insightful concerns, which we have addressed as follows:

- **Computational Cost Comparison:** We have added additional comparisons of computational cost with the baselines in Appendices C.2 and E.2 to address the concerns of reviewers $\color{red}{\text{aqgX}}$, $\color{blue}{\text{V3rz}}$, and $\color{orange}{\text{4M51}}$.

- **Comparison with More Baselines:** We have included an additional comparison with another activation steering baseline to address the concerns of reviewers $\color{green}{\text{kvZk}}$ and $\color{orange}{\text{4M51}}$.

- **More Analysis and Clarification:** We have added more details about our method in Appendix C.1 (reviewer $\color{red}{\text{aqgX}}$), further theoretical analysis in Appendix E.7.1 (reviewer $\color{red}{\text{aqgX}}$), results under adaptive attacks in Appendix E.6 (reviewer $\color{red}{\text{aqgX}}$), evaluations with different models in Appendix G (reviewer $\color{blue}{\text{V3rz}}$), results on additional benchmarks in Appendix E.5 (reviewer $\color{green}{\text{kvZk}}$), an in-depth analysis of the case study in Appendix F.2 (reviewer $\color{green}{\text{kvZk}}$), and more details about the comparison with SFT in Appendix E.2 (reviewer $\color{orange}{\text{4M51}}$).

Thank you once again for your invaluable suggestions and support.

Best regards,

The Authors

---

### Meta-Review · Area_Chair_7ek4 · 2026-01-11

**Summary:**

The paper propose a refusal steering vector (named AlphaSteer) with null-space constraint. All reviewers gave positive ratings after the author-reviewer discussion phase while initially there is one reviewer giving a negative rating (after the disucsion the reviewer upvoted). Most of the appreciation comes from clear empirical gains with theorectically supportive arguments. As all reviewers unanimously agrees to accept the submission without outstanding concerns or issues. The AC also agrees to the reviewers' appreciation to the principled method for improving LLM safety. Therefore, the AC recommends to accept the submission in ICLR 2026 with the following revision suggestions for the camera ready version:

- Null space definition may not be necessary as it is a common sense. But if the space permits, it may be okay to keep it for self-containedness.
- Explain why the name is "AlphaSteer"

**Reviewer Concerns:**

Concerns by all reviewers are well addressed by the authors' response.

**Reviewer Scores:**

All reviewers are quite active in discussion.

---

### Decision · Program_Chairs · 2026-01-26

Accept (Poster)